# Factors Associated with Job Stress among Hospital Nurses: A Meta-Correlation Analysis

**DOI:** 10.3390/ijerph19105792

**Published:** 2022-05-10

**Authors:** Ji-Young Lim, Geun-Myun Kim, Eun-Joo Kim

**Affiliations:** 1Department of Nursing, Inha University, 100 Inha-ro, Michuhol-gu, Incheon 22212, Korea; lim20712@inha.ac.kr; 2Department of Nursing, Gangneung-Wonju National University, 150 Namwonro Heungup-Myun, Wonju 26403, Korea; kimeju@gwnu.ac.kr

**Keywords:** job stress, hospital, nurses, meta-analysis, correlation

## Abstract

This study aims to investigate research trends concerning job stress among hospital nurses. Articles about job stress among hospital nurses published in English from 2008 to 2018 were searched. In the first search, 2673 articles were extracted from the MEDLINE, EMBASE, KoreaMed, KERIS, KISS, KISTI, and KMbase databases. Altogether, 154 articles were used in the systematic review and meta-analysis. Thirty-nine variables were explored regarding job stress. Among the major variables, insufficient job control, personal conflict, and burnout had a positive correlation. In contrast, intention to stay, job satisfaction, and personal accomplishment had a negative correlation. In the meta-analysis conducted in relation to a specific conceptual framework, the negative-outcome factors showed significant positive correlations with job stress, whereas the positive-outcome factors showed significant negative correlations with job stress. This study identified factors associated with job stress in nurses through a meta-correlation analysis, and the overall correlation coefficient was relatively high at 0.51. Job factors and moderators had significant meta-correlation coefficients. These results can be utilized in clinical practice and research to help develop intervention programs to relieve job stress among nurses.

## 1. Introduction

In recent years, clinical nurses have been increasingly exposed to coronavirus disease (COVID-19). Additionally, the medical environment has evolved and become specialized owing to scientific–technological advances. Such changes have led to continuously evolving job forms and higher levels of demand among hospital nurses. Moreover, pressure to provide sensitive nursing care, such as patient-centered care and activities to promote patient satisfaction, has steadily increased. This rise in job demands is provoking various levels of stress among hospital nurses. Job stress refers to stress triggered by excessive job demands that overwhelm an individual’s resources [1], and job stress in nurses refers to a state in which these overwhelming demands engender physiological, psychological, and social impairment [2].

Today, job stress has become a common and costly problem at work [3], and nursing is a stressful job due to stressors such as high expectations, excessive responsibility, and minimal authority [4]. Nurses’ job stress is a factor that lowers work productivity and efficiency, and threatens patient safety, and when job stress increases, it affects patient care as well as the quality of care [5].

Job stress in hospital nurses is influenced by various factors at the individual and organizational levels, including professional competency [6]. Changes in shift timing have a substantial impact on what is demanded of hospital nurses, which, in turn, provokes job stress, ultimately increasing organizational costs by increasing turnover and diminishing work competence [7]. For these reasons, various research attempts have been made to identify the predictors and factors associated with job stress in hospital nurses. First, job stress is influenced by individual factors, such as age, education, and experience, as proposed in the job demands–resources (JD–R) model [6,8], as well as by organizational factors, such as role ambiguity, role overload, role conflict [9], inter-professional conflict, incommensurate compensation, and authoritarian and irrational organizational culture [10].

Further, the adverse outcomes of job stress among hospital nurses include physical and mental health problems, burnout, and increased turnover intention, which, in turn, curtail work efficiency and organizational commitment, thereby impairing productivity, decreasing patient satisfaction, and hampering the advancement of the organization [7].

Factors influencing job stress among hospital nurses can be classified as antecedent factors, outcome factors, and moderator factors. By understanding the degree of relationship of these various factors with job stress, useful information can be provided for personnel management.

Ultimately, managing the job stress of hospital nurses is an essential strategy to promote work efficiency in hospitals and advance nursing organizations. In this context, many studies have attempted to identify the factors associated with job stress in nurses. However, due to the broad spectrum of these studies, more empirical findings are needed in relation to applications in clinical practice.

Therefore, this study aims to systematically review previous studies on job stress among hospital nurses and classify associated factors based on a conceptual framework in terms of a meta-correlation analysis. These findings are intended to provide foundational data for developing strategies for hospital nurses to manage job stress.

### 1.1. Purpose

This study aimed to systematically review and meta-analyze existing studies on factors associated with job stress among nurses working in general hospitals and develop nursing management strategies and intervention programs to reduce stress among them. The specific objectives were as follows:Systematically review studies on job stress among hospital nurses.Classify factors related to job stress among hospital nurses using the conceptual framework of Kath et al. [6].Examine the meta-correlation coefficients of job stress and associated factors.

### 1.2. Conceptual Framework

We used the model proposed by Kath et al. [6] as the theoretical framework. This theory classifies factors that affect job stress based on the JD–R model [8] and role stress theory [11].

This framework considers job stress in relation to stressors, outcomes, and moderators. Job stress was defined as a state perceived when work demands exceed one’s abilities [11]. The predictors of stress were categorized into individual factors, job-related factors, and hospital-related factors. Stressors and moderators refer to the factors that led to the outcomes. In this study, the moderators were defined as autonomy and leadership style [6]. Kath et al. [6] defined moderators as factors that influence the effects of stressors on outcomes. Several previous studies have demonstrated that emotional intelligence is a factor influencing job stress [12]. Emotional intelligence modulates the level of personal factors, such as hardness and resilience, according to organizational context [13]. Therefore, in this study, it was classified as a moderator.

The factors associated with job stress were assigned to each of the three categories in the theoretical framework: stressors, outcomes, and moderators. There were 19 variables classified as stressors. Individual factors included age, hardness, health-promoting behavior, personality, resilience, self-esteem, and self-efficacy. Job-related factors included emotional labor, insufficient job control, job insecurity, lack of rewards, inter-personal conflicts among co-workers, role conflict, role identity, violence, workload, and work experience. Hospital-related factors included recreation and the work environment. A total of 11 outcomes were identified, which were categorized into positive and negative outcomes. The positive outcomes included intention to stay, job performance, job satisfaction, organizational commitment, and personal accomplishment. The negative outcomes included burnout, depression, fatigue, health problems, trauma, and turnover intention. Finally, a total of nine moderators were identified, which included autonomy, competency, coping, emotional intelligence, empowerment, leadership, professionalism, self-leadership, and social support.

## 2. Materials and Methods

### 2.1. Study Design

This study is a literature review in the form of secondary data analysis and meta-analysis that aimed at identifying factors that significantly correlated with job stress among hospital nurses.

### 2.2. Data Collection 

#### 2.2.1. Inclusion and Exclusion Criteria

The literature was selected using the participant, intervention, comparisons, outcomes, time, setting, and study design (PICOTS-SD) strategies. The following studies were included: (1) studies with nurses working in a general hospital as participants (P); (2) studies with the measurement of organizational behavioral factors as the intervention (I), and (3) studies with the measurement of job stress as the outcome (O). Furthermore, the comparisons (C) were not limited because this study aimed to comprehensively explore research on measuring the variables of interest. The setting (S) was limited to the general hospital, and the study design (SD) was limited to correlation studies. Time (T) was not limited. The inclusion criteria comprised studies on hospital nurses, those that measured job stress and reported correlations between job stress and any combination of personality and job-related characteristics, and those published in Korean or English. The exclusion criteria comprised studies that did not involve healthcare staff, those that were published in other languages (excluding Korean or English), those that did not report the entire study results, and those that did not provide full manuscripts, such as conference proceedings, theses, dissertations, monographs, and books.

#### 2.2.2. Quality Appraisal

The quality of the literature was appraised using the Quality Assessment and Validity Tool for Correlational Studies, as developed by Cummings and Estabrooks [14]. This tool consists of 13 categories related to the following: application of a theoretical framework, use of randomized sampling, employment of a prospective design, appropriate sample size, data collection from multiple centers, guarantee of anonymity, response rate exceeding 60%, usage of instruments with established reliability and validity, instruments with an internal consistency of 0.7 or higher, use of appropriate statistical analyses according to the purpose of the study, and whether statistical management of outliers was presented. Studies with a score of 8 or higher for these categories were deemed as being of good quality.

#### 2.2.3. Literature Search and Selection

A literature search was performed on December 16, 2018. The search was conducted using the MEDLINE, EMBASE, KoreaMed, KERIS, KISS, KISTI, and KMbase databases. The search terms were “hospital,” “nurse,” and “job stress.” The search period was set to 10 years from 2008 to 2018. Consequently, 2673 studies were found.

After removing 1058 duplicate studies, 1615 studies were reviewed against the inclusion/exclusion criteria. After reviewing the abstracts, a total of 1355 studies were excluded: 271 studies that were not conducted on hospital nurses, 273 studies that included non-nursing staff, 221 studies that did not present job stress or other relevant factors, 18 studies that were either abstract presentations in conferences or those providing inadequate information about the study design and study results, 100 studies that were not published in Korean or English, 117 studies where the full text was unavailable, and 355 studies that did not perform correlational analysis.

The full texts of the remaining 260 studies were reviewed. A total of 26 studies were additionally excluded: five that were not conducted on hospital nurses, 15 that did not perform correlational analysis, and six that did not present the factors related to job stress. The quality of the remaining 234 studies was assessed, and 15 studies with a rating of 7 or below were excluded. As a result, 219 studies with a rating of 8 or higher were included in the descriptive analysis. Finally, 65 studies in which variables related to job stress were only reported in a single study were excluded, as meta-correlations require a minimum of two studies reporting the same variable to compute the pooled effect size. As a result, 154 studies were included in the meta-analysis (Figure 1, [13,15,16,17,18,19,20,21,22,23,24,25,26,27,28,29,30,31,32,33,34,35,36,37,38,39,40,41,42,43,44,45,46,47,48,49,50,51,52,53,54,55,56,57,58,59,60,61,62,63,64,65,66,67,68,69,70,71,72,73,74,75,76,77,78,79,80,81,82,83,84,85,86,87,88,89,90,91,92,93,94,95,96,97,98,99,100,101,102,103,104,105,106,107,108,109,110,111,112,113,114,115,116,117,118,119,120,121,122,123,124,125,126,127,128,129,130,131,132,133,134,135,136,137,138,139,140,141,142,143,144]).

Literature extraction was performed independently by two graduate students in nursing management. Cases that met the criteria were coded as 1, and those that were unsuitable were coded as 0. In case of discrepancy among the raters, whether the study was included or not was decided after discussion with the researchers.

### 2.3. Data Analysis

#### 2.3.1. General Characteristics of the Selected Studies

The following four general characteristics were examined: year of publication, country of publication, sample size, and quality rating. The results are presented in the form of frequencies and percentages.

#### 2.3.2. Classification of Variables Related to Job Stress

Variables related to job stress were classified into five factors (i.e., personal factors, job factors, hospital factors, moderators, and outcomes) based on the conceptual framework of Kath et al. [6]. The data were summarized as frequencies and percentages.

#### 2.3.3. Summary of Descriptive Statistics for Variables Related to Job Stress

The statistical significance of variables related to job stress was evaluated in terms of frequency, percentage, correlation coefficient (r), and *p*-value.

#### 2.3.4. Effect Size Calculation for Meta-Analysis and Homogeneity Testing

The correlation coefficient effect sizes of variables related to job stress among hospital nurses were statistically pooled using meta-correlation analysis (Comprehensive Meta-Analysis 3.0 software). The standardized Zr was analyzed using the standardized Fisher’s Z equation, with 95% confidence intervals. A fixed-effects model was used for high homogeneity, and a random-effects model was used for high heterogeneity. Effect sizes were interpreted per Cohen’s criteria [145]: r ≤ 0.1 (small effect size), 0.3 < r ≤ 0.5 (moderate effect size), and 0.5 < r (high effect size). Homogeneity was analyzed by computing the Q and I^2^ values following the chi-square distribution.

#### 2.3.5. Publication Bias of the Studies Included in the Meta-Analysis

Publication bias was assessed using a funnel plot, and the impact of publication bias on the results was examined via trim and fill.

### 2.4. Ethical Considerations

Ethical review by the institutional review board of the relevant university was waived for this study because it included secondary data analysis of literature (GWNUIRB-R2019).

## 3. Results

### 3.1. General Characteristics of the Studies 

A total of 154 studies covering a 10-year period (2008–2018) were located in Korean and foreign databases that explored job stress among hospital nurses through a systematic review. The largest number of studies was published in Asia (*n* = 148), which included South Korea, China, Croatia, Iran, Israel, Japan, and Taiwan. The most common sample size was 101–300 (81.82%), and the quality rating was 10 (74.31%) (Table 1).

### 3.2. Summary of Variables Associated with Job Stress

A total of 154 studies were included in the meta-analysis. Altogether, 39 variables were analyzed in relation to job stress among hospital nurses, with job satisfaction being the most studied variable (*n* = 39), followed by burnout (*n* = 38), professionalism (*n* = 29), turnover intention (*n* = 23), and work environment (*n* = 22). When classified according to the conceptual framework of Kath et al. [6], 11 of the variables associated with job stress (28.2%) fell under the outcomes category, followed by job factors (10, 25.6%) and moderators (9, 23.1%). Among the outcomes, five were positive factors (12.8%) and six were negative factors (15.4%) (Table 2).

### 3.3. Effect Size of the Variables and Homogeneity Testing

#### 3.3.1. Correlational Meta-Analysis of Job Stress and Associated Variables

Figure 2 presents the results of the meta-analysis of the correlations among major variables associated with job stress among hospital nurses. The meta coefficient for the overall correlation was 0.051 (Z = 5.08, *p* < 0.001). Positive correlations were found among the following major variables: insufficient job control (r = 0.483, Z = 8.37, *p* < 0.001), personal conflict (r = 0.454, Z = 4.96, *p* < 0.001), and burnout (r = 0.437, Z = 9.44, *p* < 0.001). In contrast, negative correlations were found among the following variables: intention to stay (r = −0.367, Z = −2.55, *p* = 0.011), job satisfaction (r = −0.311, Z = −7.01, *p* < 0.001), and personal accomplishment (r = −0.285, Z = −3.13, *p* = 0.002). The other variables did not have statistically significant correlations. The I^2^ index was 0.00~0.99.84, indicating heterogeneity; therefore, a random-effects model was used (Figure 2).

#### 3.3.2. Meta-Analysis According to the Theoretical Framework

According to the conceptual framework of Kath et al. [6], the subcategories that showed a significant positive correlation with job stress were negative outcome factors (r = 0.40, *p* < 0.001) and job factors (r = 0.29, *p* < 0.001). However, positive outcome factors (r = −0.27, *p* < 0.001), personal factors (r = −0.12, *p* = 0.024), and moderators (r = −0.11, *p* < 0.001) showed significantly negative correlations with job stress (Table 3).

### 3.4. Publication Bias in the Selected Studies

As shown in the funnel plot, no publication bias was apparent, given the symmetrical form around the integrated estimate. In trim and fill testing, there was no change before and after correction, indicating that there was no publication bias (Figure 3).

## 4. Discussion

Managing nurses’ job stress is crucial to the management and advancement of nursing organizations. This study aimed to help address issues that affect nursing organizations, such as high turnover, by identifying factors associated with job stress through a systematic review of existing literature pertinent to nurses’ job stress.

A total of 154 studies were found to have examined job stress as a study variable over a 10-year period (2008–2018), and these studies investigated various predictors of job stress. Among the 154 studies included in the analysis, 39 variables (factors, moderators, and outcomes) that had correlations with job stress were analyzed. The most extensively analyzed variables were burnout, professionalism, turnover intention, and work environment. We assigned the variables according to the model proposed by Kath et al. [6]. Most variables were assigned to the outcomes category (28.2%); positive outcomes included intention to stay, job performance, and job satisfaction, and negative outcomes included burnout, depression, fatigue, health problems, trauma, and turnover intention. In addition to the outcomes category, the variables were assigned to the moderators (23.1%), job factors (25.6%), personal factors (18.0%), and hospital factors (5.1%) categories.

A meta-analysis of variables associated with job stress among nurses showed that there was a significant overall meta-correlation of 0.051. The variables with significant positive correlations with job stress were insufficient job control, interpersonal conflict, and burnout, and those with significant negative correlations with job stress were intention to stay, job satisfaction, and personal accomplishment. Insufficient job control and interpersonal conflict are job-related factors. Pressure, role conflicts, and interpersonal conflicts at work may cause individuals to feel burdened at their job [146,147], and these are perceived as job demands that, in turn, elevate job stress. Intention to stay, job satisfaction, and personal accomplishment can be considered as outcomes of job stress, with excessive job stress likely to increase nurses’ turnover intention and diminish job satisfaction.

With reference to the model proposed by Kath et al. [6], personal factors, job factors, moderators, and outcomes were significantly correlated, but hospital factors were not. 

The results are consistent with those of previous studies [13] that reported that nurses’ self-efficacy affects job stress and has a moderating effect between job satisfaction and turnover intention.

Personal factors accounted for a relatively low rate of 18% compared to other factors. Hardiness and self-esteem were not significant. Among the personal factors, it is necessary to study the factors that can be improved by education and intervention in the future.

Job-related factors, as opposed to hospital-related factors, have the most substantial effect on job stress. This result is also consistent with the significant correlations found in relation to insufficient job control and personal conflict with job stress.

These results support previous reports that job pressure, role conflict, role identity, and inadequate rewards lead to adverse job outcomes that ultimately lead to adverse personal outcomes, such as health problems or burnout [146,147]. The JD–R model classified these factors as job demands, and although job demands are not necessarily negative, they can provoke job stress if they require further effort from individuals [8].

Moderators were negatively correlated with job stress. In the model developed by Kath et al. [6], moderators involve personal factors, job factors, and hospital factors that have an effect on the (positive/negative) outcomes of job stress. Moderators include autonomy, competency, coping, emotional intelligence, empowerment, leadership, professionalism, self-leadership, and social support. In our meta-analysis, the meta-correlation coefficients were not significant for hospital factors, but were significant for personal factors (r = −0.12, *p* = 0.024), job factors (r = 0.29, *p* < 0.001), and moderators (r = −0.11, *p* < 0.001). This suggests that nurses’ job stress is more heavily influenced by factors related to the job itself and by personal factors, as opposed to the hospital work environment, and that the factors that moderate these effects play a crucial role. In particular, autonomy, competence, emotional intelligence, and emotional labor had statistically significant meta-correlation coefficients. In other words, in addition to improving job factors to reduce job stress among nurses, efforts to prevent adverse outcomes by moderating them are equally important. Emotional intelligence has been found to be an essential leadership attribute for nursing leaders in organizational structures that feature a complex hierarchy and team systems [147]. In medical settings, emotional intelligence is an important factor in promoting multidisciplinary collaboration. Developing emotional intelligence enhances organizational commitment while reducing job stress, thereby diminishing turnover intention [148]. Positive psychological capital, such as emotional intelligence, also begets positive outcomes for organizations [149]. Autonomy lowers nurses’ job stress, promotes collaboration in the nurse–physician relationship, and increases job satisfaction [150]. Such variables with moderating effects should be taken into consideration to help mitigate job stress among nurses. Additionally, in order to alleviate the job stress of hospital nurses, nursing managers need management strategies that take into consideration factors that have a moderating effect in relation to nurses’ jobs.

This study is meaningful in that it identified various variables related to the job stress of hospital nurses through meta-correlation analysis. By subdividing and presenting these variables into nurses’ personal factors, job factors, moderators, and outcome factors, it would be easy to establish a management strategy that can reduce nurses’ job stress at the individual, organizational, and managerial levels for each factor.

This study is significant as it provides a basis for evidence-based management in preparing such management strategies.

## 5. Conclusions

In this study, we conducted a meta-correlation analysis to identify the factors associated with nurses’ job stress, and the overall correlation coefficient was relatively high, at 0.51. The main study variables were categorized into personal factors, hospital factors, job factors, moderators, and outcomes for the meta-correlation analysis. The results confirmed that job factors and moderators have significant meta-correlation coefficients. We synthesized the results pertaining to job stress, and the findings highlight the importance of considering job factors and moderators in approaches taken to alleviate nurses’ job stress. These results are likely to be useful in clinical practice and research to help develop job stress intervention programs for nurses. Despite extensive attempts, this study has limitations as it includes only studies published in Korean and English. Therefore, it is suggested that a more robust theoretical model for the factors related to job stress in hospital nurses should be developed in the future, including studies published in various languages.

## Figures and Tables

**Figure 1 ijerph-19-05792-f001:**
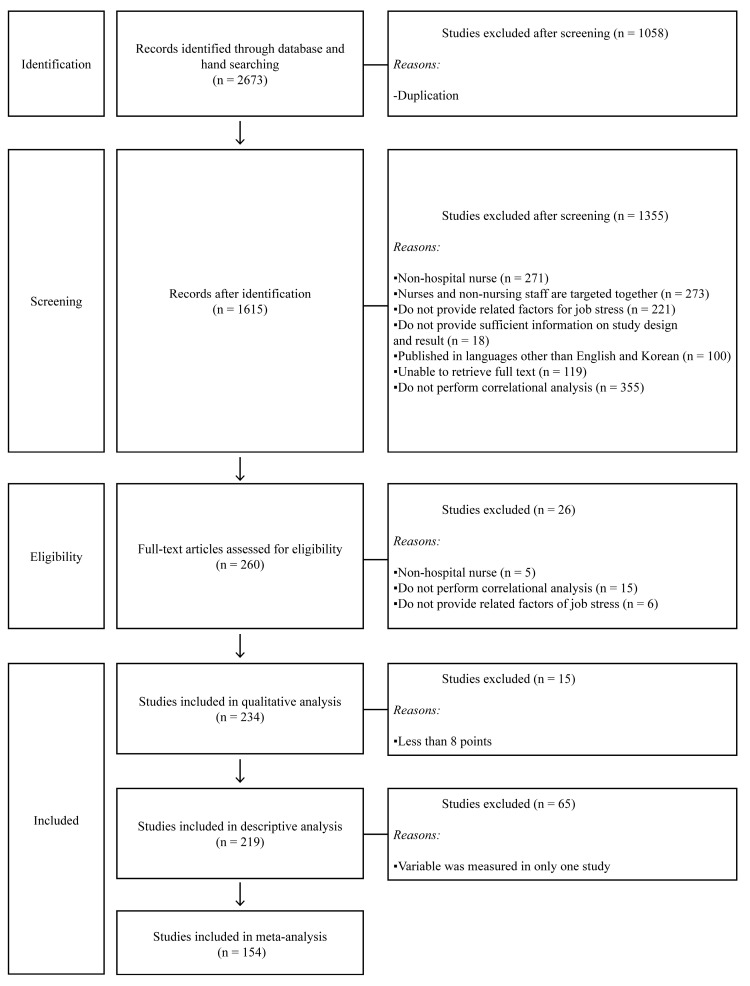
Flow diagram of study selection.

**Figure 2 ijerph-19-05792-f002:**
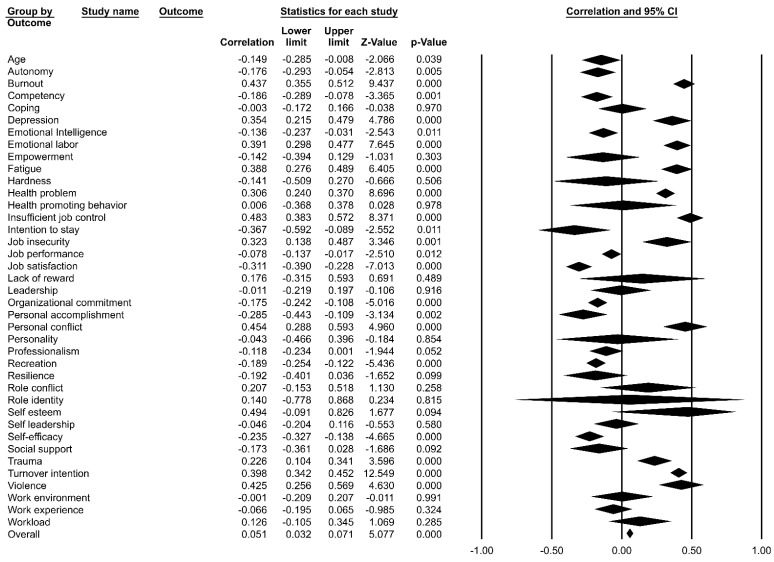
Forest plot by related variables.

**Figure 3 ijerph-19-05792-f003:**
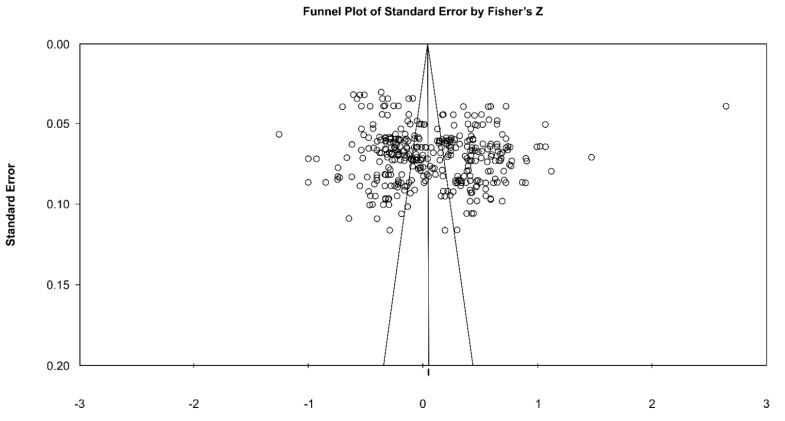
Funnel plot of the selected studies.

**Table 1 ijerph-19-05792-t001:** General characteristics (*n* = 154).

Variable	Categories	*n* (%)
Published year	2008~2010	22 (14.29)
	2011~2015	65 (42.21)
	2016~2019	67 (43.51)
Country	Africa	1 (0.65)
	Asia	148 (96.10)
	Europe	2 (1.30)
	America	3 (1.95)
Number of participants	≤100	3 (1.95)
	101~300	126 (81.82)
	301~500	14 (9.09)
	501~1000	10 (6.49)
	1001≤	1 (0.65)
Quality scoring	8	5 (3.47)
	9	32 (22.22)
	10	107 (74.31)

**Table 2 ijerph-19-05792-t002:** Classification of 39 variables based on the conceptual framework.

Category	*n* (%)	Variables (Number of Studies in Which the Variable Was Used/Studied)
Personal factors	7 (18.0)	Age (4)
		Hardness (4)
		Health-promoting behavior (9)
		Personality (4)
		Resilience (8)
		Self-esteem (2)
		Self-efficacy (17)
Job factors	10 (25.6)	Emotional labor (17)
		Insufficient job control (3)
		Job insecurity (8)
		Lack of reward (6)
		Interpersonal conflict (10)
		Role conflict (7)
		Role identity (7)
		Violence (12)
		Workload (18)
		Work experience (4)
Hospital factors	2 (5.1)	Recreation (2)
		Work environment (22)
Moderators	9 (23.1)	Autonomy (6)
		Competency (6)
		Coping (14)
		Emotional intelligence (11)
		Empowerment (4)
		Leadership (4)
		Professionalism (29)
		Self-leadership (17)
		Social support (15)
Outcomes	11 (28.2)	
Positive	5 (12.8)	Intention to stay (2)
		Job performance (2)
		Job satisfaction (39)
		Organizational commitment (14)
		Personal accomplishment (3)
Negative	6 (15.4)	Burnout (38)
		Depression (5)
		Fatigue (7)
		Health problem (9)
		Trauma (2)
		Turnover intention (23)
Total	39 (100.0)	

**Table 3 ijerph-19-05792-t003:** The meta-analysis of job stress according to the conceptual framework.

Categories		*n*	r	Lower Limit	Upper Limit	Z	*p*	I^2^
Personal factors		48	−0.12	−0.23	−0.02	−2.25	0.024	96.50
Job factors		92	0.29	0.17	0.39	4.83	<0.001	98.82
Hospital factors		24	−0.02	−0.20	0.17	−0.17	0.868	98.10
Moderators		106	−0.11	−0.16	−0.05	−3.60	<0.001	95.71
Outcomes	Negative	84	0.40	0.36	0.44	15.98	<0.001	92.10
	Positive	60	−0.27	−0.33	−0.21	−8.67	<0.001	94.00
Difference by categories			Q = 357.86		*p* < 0.001			

## Data Availability

The research data can be requested from the first author.

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
