# Peer review of "Factors Associated with Job Stress among Hospital Nurses: A Meta-Correlation Analysis"

_ijerph, 2022, doi:10.3390/ijerph19105792_

Round 1

Reviewer 1 Report

The paper is based on relevant theoretical assumptions, the complexity of which effectively requires meta-analysis work. However, I find that the theoretical background presented by the authors is too concise, as are the explanations given to the methodologies and the related discussion of the results.

Precisely because the topic is so important, a greater effort is required to study and reflect on each variable or set of variables taken into consideration.

The value of a meta-analysis consists precisely in having a general overview of all the studies carried out on a given topic and how variables and outcomes are linked to each other, in order to be able to draw conclusions on which to base a practical and theoretical reasoning. This paper, on the other hand, fails to dwell on the complexity of the subject and is limited to an exposition of variables and numbers.

I therefore recommend to greatly increase the conceptual framework, discussion and conclusion.

Finally, some other points need to be amended:

  • Please avoid the repetition of ‘whether’ in line 115-121.
  • It is not clear how the 19 stressors were identified and how the variables were assigned to the respective categories (lines 79-84). For example, why was emotional intelligence intended as a moderator although other studies in the literature consider it as a personal factor?
  • Table 2’s layout is not clear, since it is not possible to immediately understand to which category the variables listed on the right belong (are they job factors? Moderators? Etc.).
  • References list should include all the studies included in the meta-analysis.

Author Response

Response to Reviewer 1 attached below.

Reviewer 2 Report

Abstract
Initially does not contextualize the subject
Does not present the objective
does not have the problem
Does not indicate the searched databases and languages

  Introduction
Just clinical nurses?
The proposal seems to be too comprehensive for just one meta-analysis, this is not indicated!
does not have the problem
It remains to indicate the gap in the existing revisions more vigorously

Methods
The inclusion criteria are missing: Boolean operators, year of search, number of researchers, type of study analyzed
The exclusion criteria should not be the negative of the inclusion, redo.
Did they include a systematic review? Theses, dissertation, monographs?books?
Other instruments besides PICO must be eligible.

discussion
Very lean, has a lot of data and many inferences and conclusions to be discussed from the established correlations

Author Response

Response to Reviewer 2 attached below.

Reviewer 3 Report

I have completed my review of the manuscript. This topic is a very interesting and very important segment for nursing.
Introduction, purpose, conceptual framework method is very informative. The authors followed steps to conduct a systematic review and meta-analysis. According to that, this paper is suitable for publication in this journal.

Author Response

Response to Reviewer 3 attached below.

Reviewer 4 Report

There are studies in which the authors focus on reviewing the literature and conducting their research. There is nothing wrong with it. However, what should be included in this study is a broader analysis of the views of various researchers. The studies presented in the literature list are 16 items. And other studies? An extremely valuable achievement would be to go beyond the English-language studies and analyse the research results in national languages or a more significant attempt to reach American and European (European Union) data. I am convinced that such studies exist. This remark is intended further to increase the cognitive value of this work.

Author Response

Response to Reviewer 4 attached below.

Round 2

Reviewer 1 Report

The paper has been significantly improved in some essential aspects.

However, I still believe that the 'Introduction' section is too concise, while the importance of the topics would require much more articulation and attention.

This fundamental aspect, highlighted during the first round of review, was not taken into consideration.

Author Response

The manuscript has been rechecked, and appropriate changes have been made in accordance with the reviewers’ suggestions. The responses to their comments have been prepared and attached herewith. 

Thank you for your consideration.

Reviewer 4 Report

No comments

Author Response

Thank you for your consideration.